# Slope stability assessment in the seismically and landslide-prone road segment of Gerese to Belta, Rift Valley, Ethiopia

**Daniel Gebreyohannes, Ephrem Getahun, Muralitharan Jothimani** ⓘ *

Department of Geology, College of Natural and Computational Sciences, Arba Minch University, Arba Minch, Ethiopia

* muralitharan.jothimani@amu.edu.et

**Data Availability Statement:** "All relevant data are within the paper."

**Funding:** No financial support received for this study.

## Abstract

Slope instability on several sections of the Gerese-Belta route in Southern Ethiopia poses a major risk to infrastructure and safety. This research was aimed at evaluating certain areas of the road susceptible to slope instability. Through intensive fieldwork including geological analysis, surveys, and testing, three crucial slope portions were determined. Both limit equilibrium and finite element calculations demonstrated that these sections are problematic under different circumstances. The slope modification analysis shows that the safety factor increases as bench widths and the number of benches increase. In the slope section D1S3, this factor reached 1.222 when two benches measuring 5 meters in width were used on slide 2D. This initially showed an unstable safety factor of 0.26. Three benches of the same width were used under slide 2D. This resulted in a safety factor of 1.219. At the slope section (D1S2), flattening of the slope angle from initial 45˚ to 35˚, 28˚, 25˚ and 18˚ increases the factor of safety of the slope from initial 0.284 to 0.77, 0.89, 1.022, and 1.151 respectively under slide 2D analysis. At the slope section (D2S1), flattening the slope angle from initial 46˚ to 35˚, 25˚, 23˚, and 20˚ increases the safety factor from initial 0.412 to 0.684, 0.920, 1.02, and 1.315 respectively. Based on the analysis of the study results, it can be concluded that the identified slope sections are susceptible to failure under actual field scenarios, depending on the conditions under which they are predicted to occur. According to this study, the Benching method is an economical method for mitigating soil slopes, as a result of which it was recommended to be used.

## 1 Introduction

The term slope stability refers to the ability of an inclined slope to hold its own weight and external forces without experiencing any displacement, and it also refers to the potential of a naturally occurring slope or one that has been engineered to resist ground movement. Slope instability is a common and challenging problem worldwide, especially for civil engineering structures like roads, tunnels, dams, open-pit mines, and even the nature of the earth itself [1]. Ethiopia is one of the countries where slope instability is common in surfaces and bedrock in most plateau areas, including the north, south, and west [2]. Both external and internal factors

**Competing interests:** There is no competing of interest.

influence slope stability. A high incidence of slope failure and landslides is caused by external forces such as heavy rains and earthquakes worldwide, resulting in fatalities and massive economic losses [3–5].

In the event of high rainfall, agricultural activities, abnormal erosion, and seismic activity along the hill slope's gradient, hill slope instability may occur. Regarding internal factors, the slope's geometry, characteristics of the potential failure plane, and groundwater conditions emerge as paramount factors, as highlighted by [6]. The gravitational force is the principal driving force behind slope movement, directly correlating with the slope's inclination, thereby influencing slope behaviour. This influence can lead to slope instability manifesting in rotational, translational, or wedge failure modes, resulting in distinct impacts on property values, lives, and traffic obstruction, as elucidated by [7].

Multiple approaches can be employed to determine a particular slope's stability. The assessment of slope stability can be conducted through various methodologies, including Rock Mass Rating and Slope Mass Rating, as well as kinematic, deterministic, probabilistic, and numerical methods [8–10]. For rock slopes, a two-step stability assessment can be performed [11]. The first phase entails performing a kinematic analysis on the rock discontinuities to determine the underlying cause and method of possible collapse in rock slopes. The third step of the study entails the estimation of additional stability parameters. This will be accomplished by employing kinematic and limit equilibrium methods [12–15]. The objective is to ensure that the stability parameters are aligned with the specific type and mechanism of rock slope failure discovered through the kinematic evaluation [16–19]. Historically, slope stability analysis predominantly relied on more straightforward calculations, facilitating slope stability evaluation. However, in contemporary times, the availability of powerful computer software has become increasingly prevalent. Consequently, experts have developed more intricate yet precise methods for slope stability analysis.

A limit equilibrium analysis is the most commonly used method for assessing slope stability. It serves to identify potential failure mechanisms and calculate the safety factor for a given geotechnical situation within the method's constraints [20–25]. According to [26], limit equilibrium methods are relatively straightforward compared to numerical approaches [26]. For slope stability analysis, the finite element method presents a compelling alternative to traditional methods due to its precision, adaptability, and reduced need for a priori assumptions, especially regarding failure mechanisms. According to the finite element model, slope failure occurs when the soil lacks adequate shear strength in regions subjected to shear stresses, leading to soil failure. The FEM technique offers the capability to perform robust engineering calculations, utilizing modern computational tools to replicate the physical characteristics of a problem without simplification [27–29].

The Finite Element Method (FEM) offers several advantages. For instance, it allows for the analysis of slope geometry in two and three dimensions, facilitates the examination of linear and nonlinear problems, and computes Factors of Safety (FOS) without requiring assumptions about the shape or position of the sliding plane. Additionally, the FEM method can accommodate various soil material models and consider the stress-strain relationship of the materials, rendering it a more suitable choice compared to the traditional limit equilibrium method [30]. Furthermore, FEM can furnish insights into deformations at different stress levels, enabling the monitoring of progressive failure, including overall shear failure. This feature enhances the realism of the analysis results, setting them apart from those obtained through a limit equilibrium analysis [28].

During the construction of the road segment connecting Gerese town to Belta, the area under current investigation, issues related to slope instability have been frequently observed. These problems tend to exacerbate during the rainy season, resulting in various rock slope

failures, including planar, wedge and rock slope collapses, encroaching upon the road and damaging infrastructure. Additionally, a section of the road has been affected by soil slope failures, characterized by nearly circular shapes, rendering it almost impassable. These slope stability challenges have not only obstructed the road but have also inflicted damage on it, disrupting daily traffic flow. No prior efforts had been made to assess or address the slope stability issues along this road section—neither through investigation nor remediation measures. Therefore, the primary aim of this study was to systematically identify, locate, and evaluate slope stability concerns along specific road segments. Various approaches were employed to achieve this, including deterministic and probabilistic methods. The ultimate objective was to propose appropriate remedial measures based on the analysis findings. This study was conducted with a set of following specific objectives.

➢ Identification and localization of potential slope instability issues: This study's primary objective was to identify and precisely locate areas along selected road sections that displayed signs of potential slope instability. The process involved a comprehensive field assessment and geological investigation to determine vulnerable areas.

➢ Slope stability analysis utilizing limit equilibrium and finite element methods: The second objective was to conduct an in-depth slope stability analysis within the identified problem areas. A comprehensive understanding of stability conditions and potential failure mechanisms was achieved through the rigorous application of both the limit equilibrium and finite element methods.

➢ Identification of causative or triggering factors for slope instability: This research aimed to identify the underlying causes or triggering factors that contributed to slope instability in the study areas. An extensive review of geological, geotechnical, and environmental factors, as well as historical data and seismic activity records, was involved in this process.

➢ Recommendation of potential remedial measures: The final objective of the slope stability analysis was to propose a range of viable and effective remedial measures. Based on a comprehensive understanding of the geological and geotechnical conditions, these recommendations encompass engineering solutions, mitigation strategies, and land use management practices to enhance slope stability.

The novelty of this study lies in its comprehensive approach to addressing slope instability issues in the Gerese to Belta region, Rift Valley, Ethiopia. The novelty can be summarized as follows:

➢ Precise localization of vulnerable areas: This study innovatively pinpointed and precisely localized areas susceptible to slope instability. As a result of combining extensive field assessments with geological investigations, it provided a higher level of accuracy in identifying problematic zones compared to conventional estimates.

➢ Integration of multiple analytical methods: The study introduced a novel approach by integrating the limit equilibrium and finite element methods in the slope stability analysis. In addition to improving the accuracy of the assessment, this comprehensive analytical framework offers a deeper understanding of stability conditions and potential failure mechanisms.

➢ Holistic causative factor identification: In terms of causative factors, this research pioneered a holistic approach by considering a wide array of geological, geotechnical, environmental, historical, and seismic characteristics. As a result of this multifaceted analysis, slope instability triggers have been understood more comprehensively and nuanced.

## 2 Materials and methods

### 2.1 Study area description

The road is located in the southern part of the country in the Southern Nations and Nationalities People Regional State. It starts at a village called Shele Mazoria, 20 from Arbaminch on the Arba Minch–Jinka road and connects Arba Minch to Sawla via Kemba. The study area for this research work is located in North Omo, Southern Ethiopia (Fig 1). It is situated around 489 Km from the capital, Addis Ababa, in the Southeastern direction. Gerese–Belta road is extended between 5˚ 55' 00" N, 37˚ 18' 17" E to 6˚ 03' 23" N, 37˚ 16' 31" E, which connects the town Gerese and Belta. The boundary area of the road is routed by one kilometer. Both sides take two kilometres and three slope sections from the road. The National Meteorological Service Agency records from the Gerese Observatory Sub-station show that the mean annual rainfall for the past eleven years (2012–2022) is 184.5mm at an altitude of 2,402 m above mean sea level. High precipitation occurs in April, May, and October, covering around 50% of the annual rainfall.

### 2.2 Seismicity of the study area

As per the Ethiopian Building Code of Standards, Ethiopia is divided into five primary earthquake zones, ranging from zones with no risk of damage (zone 0) to zones with substantial potential for damage (zone 3 and zone 4). The study location falls within the Rift Valley region, as shown in the seismic hazard map in Fig 2. It is located in an earthquake zone with Modified Mercalli intensities ranging from VI to IX [31]. During slope analysis, it is imperative to consider seismic loading and mitigate its effects on safety. Consequently, during the initial design phase, it is advisable to consider a minimum horizontal seismic acceleration of 0.1g following

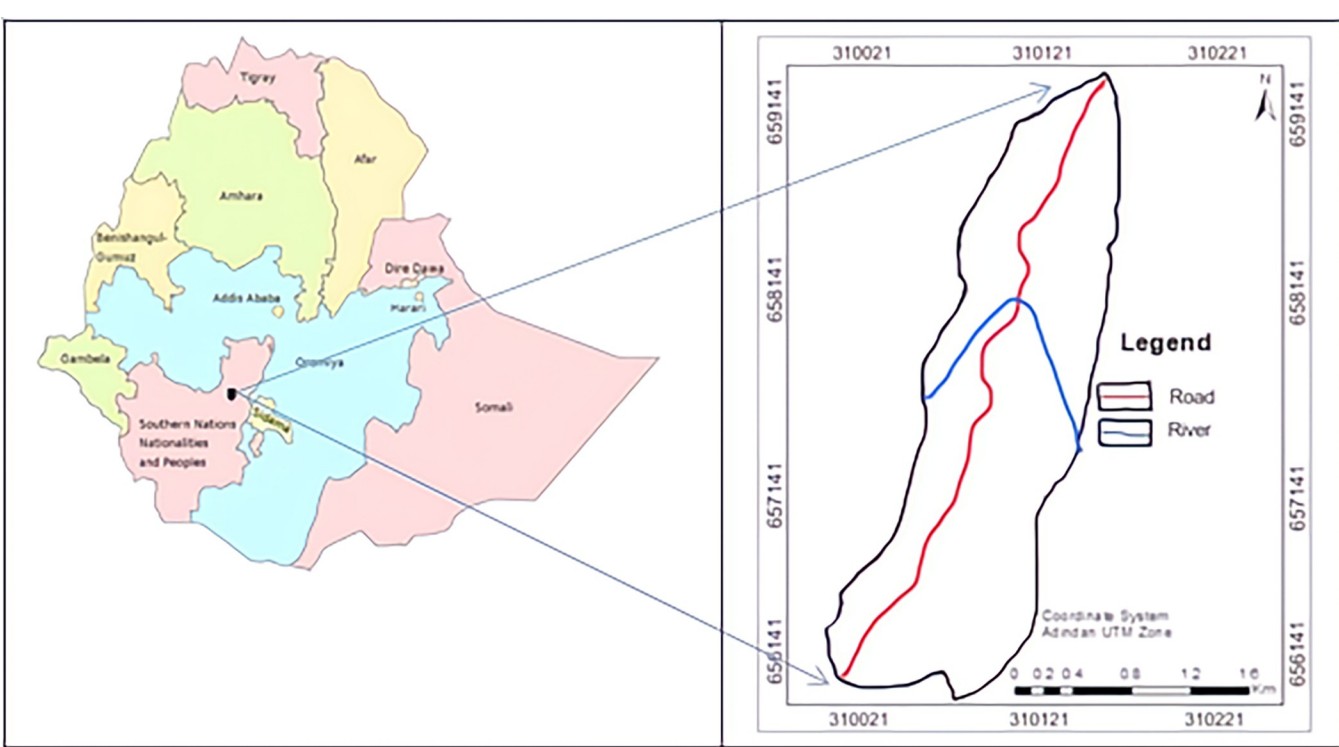

**Fig 1. Location map of the study area (Drawn with QGIS software).**

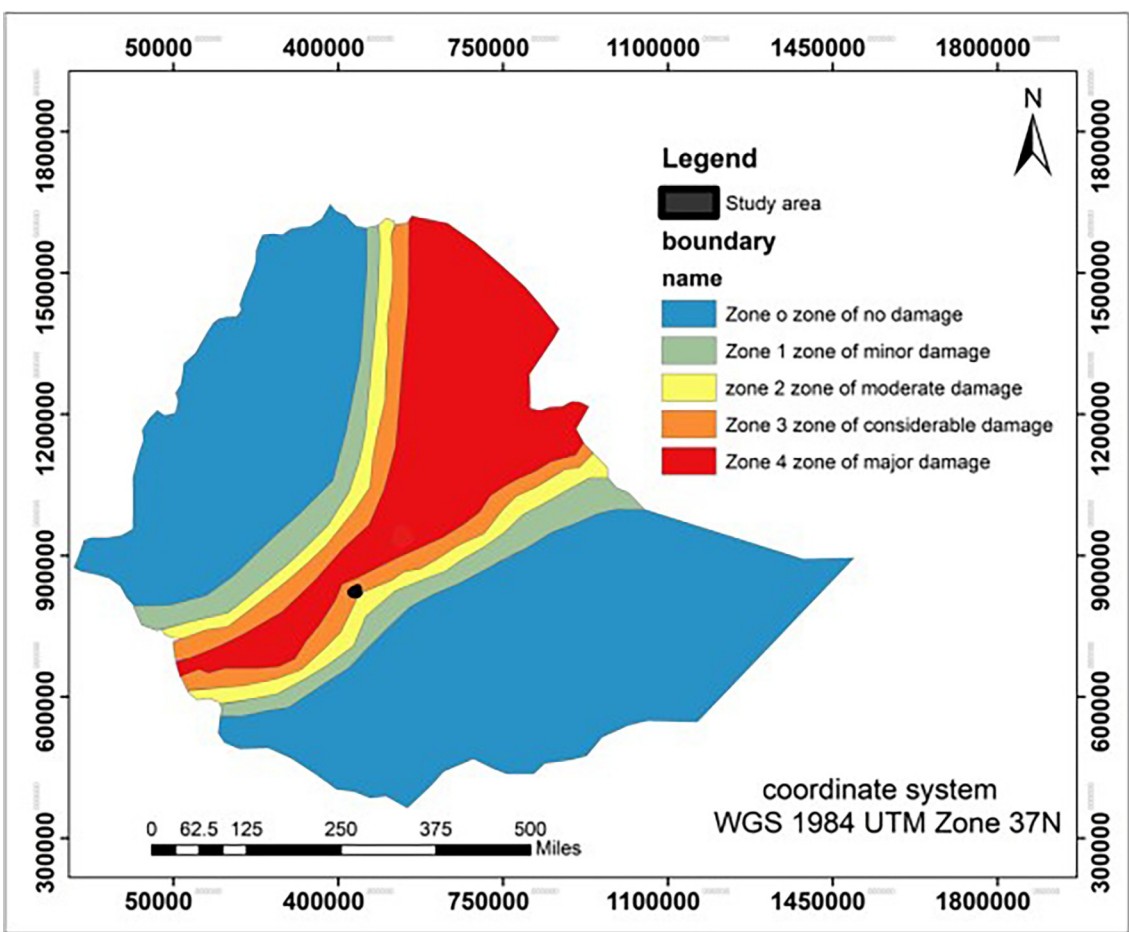

**Fig 2. Seismic hazard map of Ethiopia for 100 years return period modified after [31].**

Ethiopia's seismic hazard zones, which are based on anticipated ground acceleration amplitudes over 100 years.

## 2.3 Materials

A combination of laboratory and field materials and equipment was used in the research. During fieldwork, GPS devices, compasses, and geological hammers were used to collect, measure, and test data. Soil mechanics laboratory equipment like direct shear test apparatus, balance, beakers, and cylinders were used for lab works of shear strength tests and unit weight tests. The data processing interpretation was done with the help of different software such as Arc GIS, Microsoft Excel, slide 6.00, Slide 2D, and Phase 2D for slope stability analysis.

## 2.4 Methods

**2.4.1 Data sources.** Secondary data was provided by the Ethiopia Road Authority (ERA), the National Meteorological Service Agency (rainfall and groundwater depth data), and the Geological Survey of Ethiopia. Using Microsoft Excel and data obtained from the Ethiopian National Meteorological Service Agency's Gerese station, a graph was created showing the mean monthly rainfall for the study area. In addition to providing valuable information regarding geological formations, types of lithology, and slope groundwater conditions, the

field investigation offered helpful information. Several observable field characteristics were also used to identify critical indicators of slope instability, such as removing the slope's base, the appearance of tension cracks, and tilting the slope's face. To map surface structures such as joint sets, the Brunton compass was used to determine their orientation (including dip and dip directions) concerning slopes and discontinuities.

**2.4.2 Laboratory test analysis.** A representative soil sample was extracted from the test pit to conduct stability analyses on the soil slope and determine the shear strength parameters and unit weight. Both unit weight and direct shear strength tests were conducted on soil samples collected at depths ranging from 1 meter to 2 meters.

*2.4.2.1 Direct shear test.* The direct shear test is employed in this study to assess the shear strength properties of soil samples. As these soil samples are obtained from critical sections of roads, they are undergoing this testing procedure following [32] guidelines. Direct shear tests are fundamental geotechnical experiments used to understand how soil materials respond to shear forces. During the construction and maintenance of roads, it is imperative because it determines the soil's ability to withstand lateral stress and deformation, which is essential to ensuring the stability and durability of the road structure.

As part of the procedure, representative soil samples are collected from specific locations along the roadway that are deemed critical. A critical section is typically selected according to soil composition, moisture content, and geological conditions, significantly impacting the road's performance and safety. After collecting soil samples, they are carefully prepared and placed within the direct shear apparatus, which consists of two halves of a shear box. A soil sample is placed between these halves, and an average load is applied to simulate the vertical stress that the soil experiences in its natural environment. The soil sample is then sheared along a predefined plane using a horizontal force.

During the test, the applied shear force is gradually increased until the soil sample reaches a critical state and begins to deform or fail. It is precisely monitored at this crucial point and the corresponding shear stress and displacement data. These data are analyzed to determine various parameters, including the soil's shear strength, cohesion, and angle of internal friction. The direct shear test provides valuable information that can be used to inform road design, construction, and maintenance decisions. By assessing the shear strength of soil samples from critical road sections, engineers can make informed decisions about the type and thickness of road materials, slopes, and stability of road embankments, ensuring the road infrastructure's safety and longevity. Further, adherence to standard testing procedures, such as [32], provides the consistency and reliability of the obtained shear strength data, making it an invaluable tool for geotechnical engineers.

*2.4.2.2 Unit weight test.* The soil unit weight test was conducted using representative samples collected during an extensive field survey. During the testing, both bulk unit weight and dry unit weight measurements were obtained. Based on the bulk unit weight, the dry unit weight was calculated. Table 1 summarizes the laboratory soil tests conducted in this study.

*2.4.2.3 Data processing and analyses.* The collected data from secondary and primary sources through desk study, reconnaissance and detailed field survey, laboratory test, and literature review were organized and processed so that they can be easily accessed during further

**Table 1. Summary of laboratory tests on soil samples.**

| Laboratory test | Type of Material | No of sample | No of trial |
|---|---|---|---|
| Unit weight test | Soil | 2 | 2 for each |
| Direct shear test | Soil | 2 | 3 for each |

stability analyses. After the whole data needed for stability analyses was collected and processed, the stability of the slope was analyzed by determining the factor of safety based on limit equilibrium and finite element methods for identifying critical soil slopes.

*2.4.2.4 Limit equilibrium method.* The limit equilibrium method is widely used to assess the stability of slopes in two and three dimensions. As described by [33–35], this method involves identifying potential failure mechanisms and calculating factors of safety specific to a given geotechnical scenario. The factor of safety for critical soil slope sections was computed using the slide software, employing the Spencer method from among the various Limit Equilibrium Methods (LEM). Based on the fact that the Spencer method yielded a lower safety factor than other methods, it was selected.

Shear Strength Equations: The LEM uses equations to calculate the resisting force based on shear strength parameters. For example, the Mohr-Coulomb equation for shear strength is employed as follows the Eq 1:

$$T = c' + \sigma'*\tan(\varphi') \qquad (1)$$

Where, T (Shear strength) = c' (Effective cohesion) + σ' (Effective normal stress) * tan(φ') (Effective friction angle)

Driving Force Equations: The driving force can be calculated based on the weight of the soil mass as follows the Eq 2:

$$D = \gamma*H*L \qquad (2)$$

Where D (Driving force) = γ (Unit weight of the soil) * H (Height or depth of the soil mass) * L (Width of the potential failure surface)

Factor of Safety Equation: The FoS is calculated by equating the resisting force to the driving force as follows the Eq 3:

$$FoS = (c' + \sigma'*\tan(\varphi'))/(\gamma*H*L) \qquad (3)$$

Where FoS (Factor of Safety) = (c' (Effective cohesion) + σ' (Effective normal stress) * tan (φ') (Effective friction angle)) / (γ (Unit weight of the soil) * H (Height or depth of the soil mass) * L (Width of the potential failure surface).

Slope Stability Analysis: In slope stability analysis, the critical sliding surface is determined by considering various potential failure planes within the soil mass. The FoS is computed for each plane, and the most critical one is selected.

Retaining Wall Stability: For retaining walls, the LEM assesses the stability by considering the forces acting on the wall, including the weight of the soil behind the wall and any surcharge loads. The FoS is calculated to ensure the wall can resist these forces.

*2.4.2.5 Finite element method.* With the advancement of computer technology, the finite element method has emerged as an alternative to conventional methods in geotechnical engineering. In the context of slope stability analysis using the finite element method, reliance is placed on the shear strength reduction (SSR) technique, which proves beneficial in determining the factor of safety and assessing improvements post-strengthening, according to [20, 28]. Using Phase 2D software, the Factor of Safety was determined using the Finite Element Method (FEM) of slope stability analysis. The Mohr–Coulomb model was used to calculate the safety factor for three different soil slopes under various anticipated conditions, including static dry, dynamic dry, static wet, and dynamic wet scenarios.

# 3 Results and discussions

## 3.1 Geology of the area

A comprehensive field survey was conducted to identify and characterize geological materials, such as soils, on the steep sections of the slope and to identify the most vital areas of the slope. Three critical soil slope failures were identified using visual cues, such as scarp faces and areas where the soil had collapsed. Structural factors did not influence these failures. Furthermore, the slope was visually observed to be saturated, indicating the presence of moisture. The groundwater status of the study area was indirectly assessed through on-site observations of water stains and springs on the slope's surface. In particular, springs and other water features were observed near the critical slope sections. According to Ethiopian Meteorological Agency data, the groundwater is also deep, around 3–10 meters. So, groundwater can be considered the main destabilizing factor for this specific area, and rainfall can be regarded as a destabilizing factor. Table 2 shows the location and descriptions of critical slope sections identified during the detailed field survey. The following is a description of the types of soils and rocks found in the study area:

- There were top soils, which were reddish, that covered the top part of an area that was deep down up to two meters. There were reddish rhyolite rocks and slightly weathered, highly weathered fractured rhyolite that covered a significant part of the study area.

- Colluvial soils are composed of detritus materials from soil horizons and parent materials from slopes of slopes transported locally by landslides or water erosion.

- In addition to the above, residual soils formed due to weathering and decomposition of rocks that had not been transported from their original place. This study area has a residual soil of ignimbrite rock (reddish to brown) and pumice rock (light grey to medium gray).

## 3.2 Results of laboratory analysis

Various soil tests were conducted in this study to assess engineering properties, particularly the shear strength, which denotes the maximum resistance to shear stresses just before soil failure. A soil's shear strength is the primary engineering characteristic that determines soil's stability under loads, slopes' stability, and the exerted earth pressure against retaining structures. The shear failure of a soil mass occurs when the stress induced by applied compressive loads exceeds its shear strength [36]. The soil tests that have been conducted for this study are the following:

**3.2.1 Shear strength test.**   Direct shear testing is one of several methods employed to determine the shear strength of soil, specifically the maximum resistance of soil to shear stresses just before soil failure occurs. Using soil samples (D1S2 and D1S3) extracted from test

**Table 2. Location and descriptions of critical slope sections identified during detailed field survey.**

| Slope sections | Northing | Easting | Characteristics of the slope sections are based on the field manifestations. |
|---|---|---|---|
| D1S2 (Soil) | 657141 | 310021 | Critical soil slope with larger scarp face, nearly circular and composed of colluvial soil. |
| D2S1 (Soil) | 653349 | 313714 | Critical soil slope with larger scarp face of residual soil. |
| D1S3 (Soil) | 650692 | 315727 | Critical soil slope with moderately steep-sloped sandy soil that originated from parent material Pumice rocks (light grey to medium gray). |

D = Day S = Station

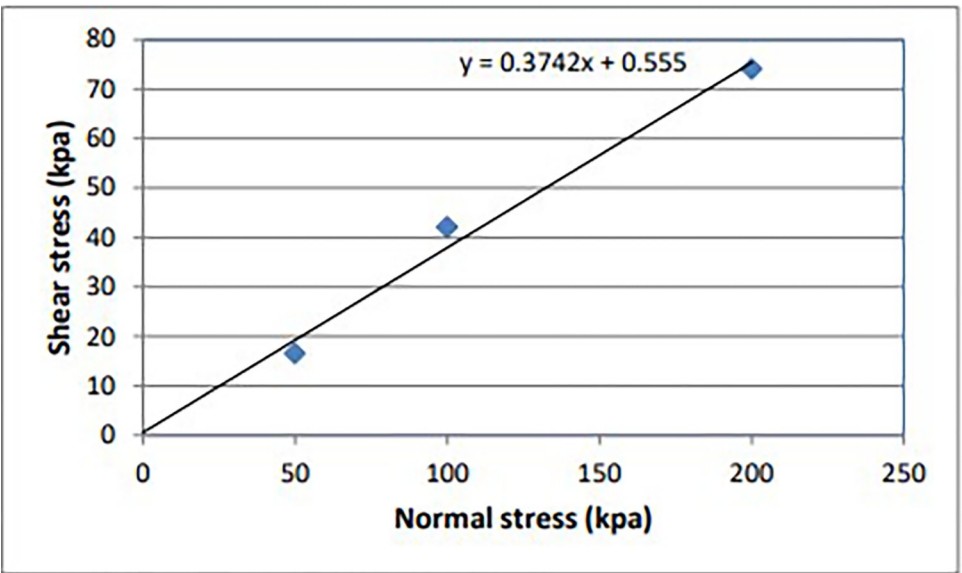

**Fig 3. Normal vs shear stress graph of direct shear test of soil sample (D1S3).**

pits at depths ranging from 1 meter to 2 meters, a direct shear test was conducted following [37]. Three trials were conducted, with normal stresses of 50 kPa, 100 kPa, and 200 kPa applied in each trial, with the stress levels increasing progressively from the first to the third trial. Test results have been tabulated and graphically presented below. The shear stress was calculated from the following formula: the shear force and area utilized in the test.

$$\text{Shear stress (kpa)} = \text{Shear force/area} * 1000$$

Then, using maximum shear stress (kpa) and normal stress (kpa) from the test of soil samples of different trials, the graph of shear stress vs. normal stress was plotted to determine cohesion and friction angle (Figs 3 and 4).

A direct shear test conducted on the soil provided crucial parameters, such as the cohesion value and friction angle. Subsequently, these parameters were used as input variables for soil stability analysis.

**3.2.2 *Unit weight test*.** The unit weight test of soil was conducted at the laboratory on the representative soil samples collected from two critical soil slope sections. Below Table 3 shows the summary of the unit weight test of the soil sample.

**3.2.3 Soil slope stability analysis based on limit equilibrium.** This study incorporated the soil slope stability analysis and measurements of the lithology dimensions in critical soil slope sections into Slide 2D software as a polygon input. Slide 2D software was utilized to assess the stability of the slope section under static and dynamic loading conditions, including analyses for saturated and dry conditions. This evaluation was conducted using the Spencer method, a Limit Equilibrium Method (LEM) component. Table 4 shows the parameters for input into slide 2D for soil slope.

According to the analysis results, the soil slope section (D1S2) is unstable under all expected scenarios. The result of analysis under slide 2D software indicates that the soil slope is highly susceptible to failure even if under dry, static conditions without an effect of external factors or loadings like seismic loading and groundwater effect. In addition to this, since an area is situated in a seismically active zone, the result would be severe as an earthquake happens suddenly and heavy rainfall occurs.

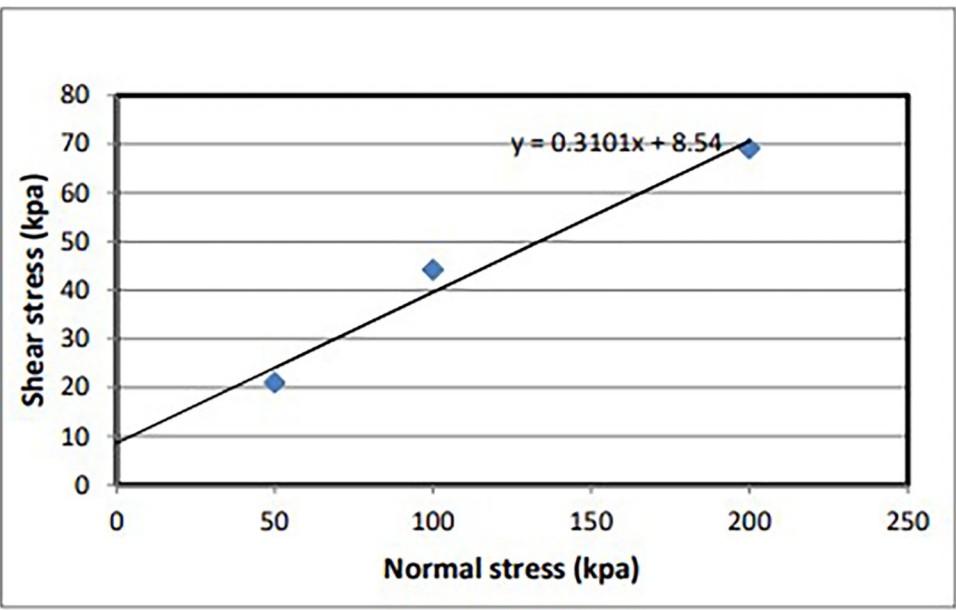

**Fig 4. Normal vs shear stress graph of direct shear test of soil sample (D1S2).**

**Table 3. Summary of unit weight test of soil sample collected from two critical sections.**

| Soil type | Sample no | Ms total (g) | Ms dry (g) | Vs ($cm^3$) | ρ = M/V g/ $cm^3$ | γsat = ρ. $KN / M^3$ | γdry = ρ. s $KN / M^3$ |
|---|---|---|---|---|---|---|---|
| Pumice | 1 | 29.33 | 29.28 | 22.21 | 1.32 | 13.2 | 13.2 |
| Residual | 2 | 24.27 | 22.61 | 12.1 | 2.006 | 19.66 | 18.7 |

**Table 4. Parameters for input into slide 2D for soil slope.**

| Soil slope sections | Unit weight ($KN / M^3$) | | Cohesion in (kpa) | Friction angle(˚) | Seismic coefficient | Slope angle in (˚) | |
|---|---|---|---|---|---|---|---|
| | γsat | γdry | | | | | |
| D1S2 | 19.66 | 18.7 | 8.54 | 17.22 | 0.1 | Top | 20 |
| | | | | | | Bottom | 45 |
| D1S3 | 12.9 | 12 | 0.6 | 20.516 | 0.1 | Top | 65 |
| | | | | | | Bottom | 40 |
| D2S1 | 19.66 | 18.7 | 8.54 | 17.22 | 0.1 | Top | 75 |
| | | | | | | Bottom | 46 |

Based on the Limit Equilibrium analysis presented in Slide 2D, the soil slope segment labelled D1S3 is also classified as unstable. The factor of safety is below one in all foreseeable scenarios. According to the slope stability analysis conducted in Slide 2D, D1S3 exhibits a notably high susceptibility to slope instability compared to other unstable slope sections. It has an extremely low factor of safety (specifically, 0.264, 0.156, 0.122, 0.096) in dry static, dry dynamic, static wet, and dynamic wet conditions, respectively. Therefore, the D1S3 slope section is highly susceptible to slope failure.

Also, the soil slope segment labelled D2S1 is characterized as unstable, with its safety factor below one in all expected scenarios. The introduction of water to these soil slope sections has a significant impact on their stability. The factor of safety decreases from 0.545 under static dry

**Table 5. The factor of safety determined to limit the equilibrium method of soil slope sections.**

| Slope sections | Loading conditions | | | |
|---|---|---|---|---|
| | Static | Strength Reduction Factor | Dynamic | Factor of Safety |
| Soil slope section (D1S2) | Dry | 0.487 | Dry | 0.371 |
| | Wet | 0.284 | Wet | 0.240 |
| Soil slope section (D1S3) | Dry | 0.267 | Dry | 0.156 |
| | Wet | 0.122 | Wet | 0.096 |
| Soil slope section (D2S1) | Dry | 0.545 | Dry | 0.412 |
| | Wet | 0.284 | Wet | 0.252 |

conditions to 0.284 under static wet conditions, indicating that the slope's stability is influenced by pore pressure, which may be induced by water. Table 5 provides a comprehensive summary of the factor of safety calculations using the Spencer method in Slide software.

**3.2.4 Finite element method slope stability analysis of soil.** The Finite Element slope stability analysis method assessed three critical soil slope sections using Phase 2D. It determined the factor of safety under static and dynamic loading conditions in both dry and wet conditions. In this study, the dynamic analysis of the slope sections incorporated the utilization of the horizontal earthquake coefficient. Next, the Gaussian Elimination solver type was employed to generate the input parameter, which served as an input value for both the slope section and the mesh. The Mohr-Coulomb model (MC) was used for analysis. Input parameters for the three soil slope sections used in this model are Unit weight, Cohesion (c), Friction angle (φ), Dilatancy angles (ψ), Young's modulus (E), Poisson's ratio (v). These parameters, such as strength and stiffness, are determined using data collected from fieldwork and laboratory tests.

For this analysis to determine the safety factor using Phase 2D software, the parameters used as input parameters are presented in Table 6 below. Phase 2D software examined all three soil slope sections in static, static, dynamic, and dynamic wet conditions.

According to the analysis results, the safety factor for slope section D1S2 is consistently below one, indicating instability under all expected conditions. Additionally, in both static and dynamic scenarios, the factor of safety diminishes with the transition from dry to wet conditions. Clearly, water has a substantial influence on this precarious soil slope section. In addition to water's impact, seismic activity is also considered when evaluating this slope section's stability. In these soil slope sections; the safety factor is very small according to phase 2D analysis that reveals the slope section of soil is highly unstable and almost ready to fail. Additionally, the value of the factor of safety reduced from dry to wet and static to dynamic conditions reveals that the total effect of groundwater and earthquake enhances the vulnerability of slope sections to failure.

In soil slope section D2S1, the analysis results show that the slope section's safety factor is less than one at all anticipated conditions as it was under Slide 2D software. In addition, the

**Table 6. Input parameters of phase 2D software used under numerical modeling of soil slope section.**

| Soil slope section | Unit weight (KN / $M^3$) | | Cohesion In $KN / M^2$ | Φ (˚) | (ψ˚) | Poisson ratio (v) | E (kN/m2) |
|---|---|---|---|---|---|---|---|
| | γsat | γdry | | | | | |
| (D1S2) S1 | 19.66 | 18.7 | 8.54 | 17.22 | 0 | 0.163 | 15600 |
| (D1S3) S2 | 12.9 | 12 | 0.6 | 20.52 | 0 | 0.15 | 12000 |
| (D4S1) S3 | 19.66 | 18.7 | 8.54 | 17.22 | 0 | 0.163 | 15600 |

Where (φ)- Friction angle (ψ)—Dilatancy angles, (E)—Young's modules

**Table 7.  Summary of strength reduction factor value determined under finite element method.**

| Slope sections | Loading conditions | | | |
|---|---|---|---|---|
| | Static | Strength Reduction Factor | Dynamic | Strength Reduction Factor |
| Soil slope section (D1S2) | Dry | 0.40 | Dry | 0.30 |
| | Wet | 0.22 | Wet | 0.16 |
| Soil slope section (D1S3) | Dry | 0.21 | Dry | 0.12 |
| | Wet | 0.01 | Wet | 0.00 |
| Soil slope section (D2S1) | Dry | 0.50 | Dry | 0.40 |
| | Wet | 0.24 | Wet | 0.20 |

value of the factor of safety reduced from dry to wet conditions as well as static to dynamic situations. This shows that additional external factors, like the introduction of water and the earthquake loading process, could exaggerate the instability of the slope section. Generally, at all three unstable soil slope sections, the safety value calculated under Phase 2D software is closer to the value of safety calculated under Slide 2D software. The detailed summary of the factor of safety calculated under Slide software using the Spencer method is presented in Table 7 below.

## 3.3 Comparing of results of the Limit Equilibrium Method (LEM) and Finite Element Method (FEM)

Compared the limit equilibrium and finite element slope stability analysis methods based on the safety strength reduction factor results from both Slide 2D and Phase 2D software. In assessing the factor of safety values generated by the Slide software, we find that they are slightly higher than those produced by the Phase 2D finite element methods. Phase 2D provides a more comprehensive understanding of stress, strain, deformation, and displacement at any location within the slope section compared to Slide 2D. The Finite Element Method (FEM) can also monitor progressive failures, including the possibility of overall failure due to shear. Accordingly, Phase 2D results are considered more reliable than those derived from Slide software. When applied in the limit equilibrium method, the Slide 2D software yields a uniform safety factor exclusively along the failure surface.

## 3.4 Remedial measures

Among the various slope stabilization techniques, a geometric approach is straightforward and cost-effective, although sufficient space is necessary for its implementation. Slope stability can be improved by reducing steep slopes to mild gradients. By excavating the slope, excess loading can be eliminated. Using Slide 2D, we examined the use of benching and flattening of slope techniques to stabilize unstable soil slopes. In order to determine the most secure stability conditions for slope segments, we evaluated the impact of reducing the slope angle and implementing benching. For stabilizing this slope, the benching method is the most cost-effective option. In addition, we proposed other methods that were aligned with specific circumstances.

**3.4.1 Benching technique analysis using slide 2D.**   Benching is an effective method for enhancing slope stability by dividing the slope into several segments, thereby reducing the likelihood of failure. According to Slide 2D, among the three unstable soil slopes examined, slope section D1S3 was explicitly used to determine the optimal slope design by utilizing the benching technique within the stabilization method. A static dry condition with a safety factor of 0.267 was selected for this analysis.

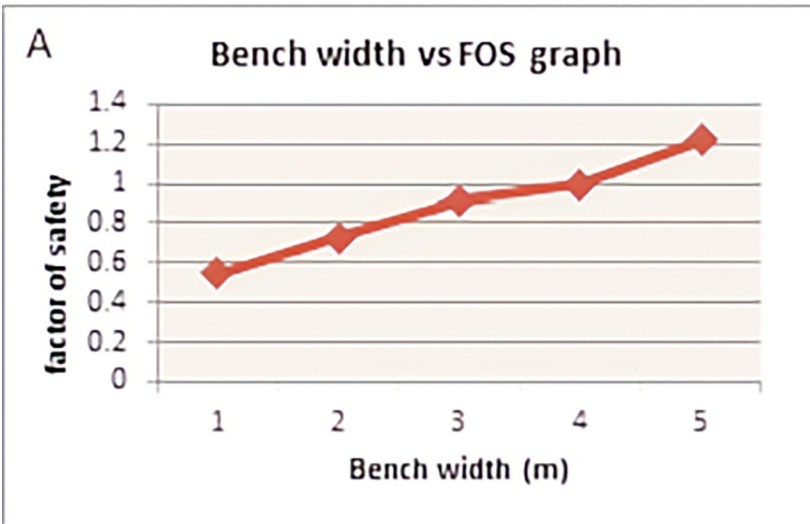

| Bench width(m) | FOS from slide 2D |
| --- | --- |
| 1 | 0.547 |
| 2 | 0.725 |
| 3 | 0.919 |
| 4 | 1.001 |
| 5 | 1.222 |

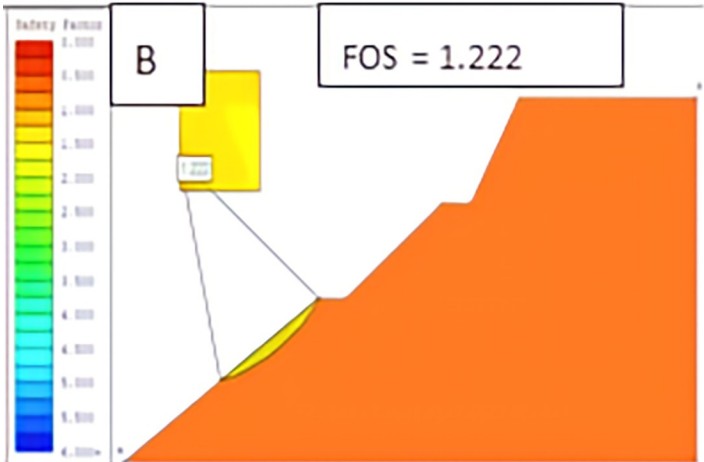

**Fig 5. The effect of bench width on a factor of safety (D1S3).**

Using Slide 2D software, we examined the impact of bench width and bench number on the stability of the slope cut. This was achieved by varying the bench width while maintaining the slope height for the same slope section and increasing the number of benches within the same width. Figs 5 and 6 illustrate that as the bench width increased from 1 meter to 5 meters, the safety factor increased from 0.547 to 1.222. The safety factor increased from 0.794 to 1.219 as the number of benches increased from one to three with a constant bench width of three meters. So, the slope became stable as the width of the bench reached 5m, and the number of benches reached 3 under another analysis. The analysis of bench width and bench number is presented below in the figure separately. Benches were used on both top and bottom slopes in this slope section. The Spencer method calculated the safety factor as the former analysis.

**3.4.2 Using 2D slide software to flatten slope angles.** Compared to the gentler slope segment, the steeper slope segment is more prone to sliding due to the force of gravity. Based on Slide 2D analysis, the slope angle's impact was assessed under the most unfavourable conditions, with a factor of safety of 0.240 for the dynamic saturated soil slope section (D1S2). According to the analysis shown below (Fig 7), the factor of safety, which was initially 0.284 at

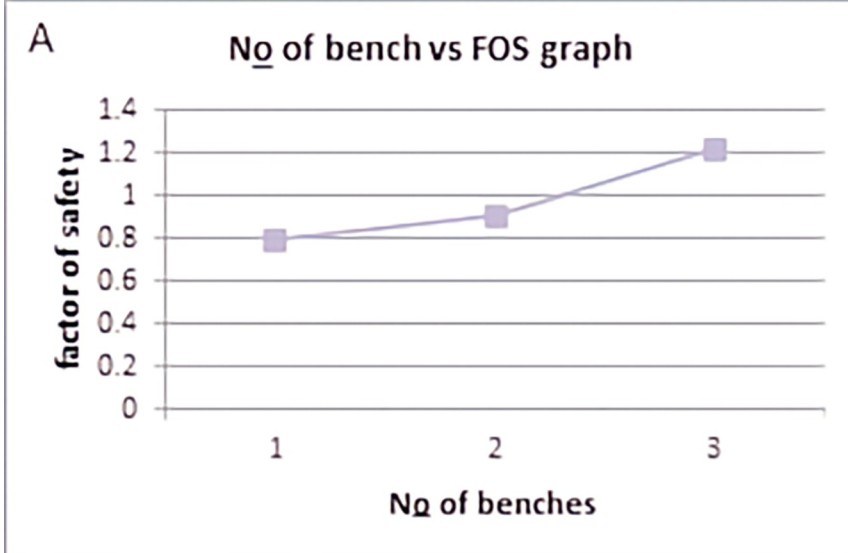

| No of benches | FOS from slide 2D |
|---|---|
| 1 | 0.794 |
| 2 | 0.902 |
| 3 | 1.219 |

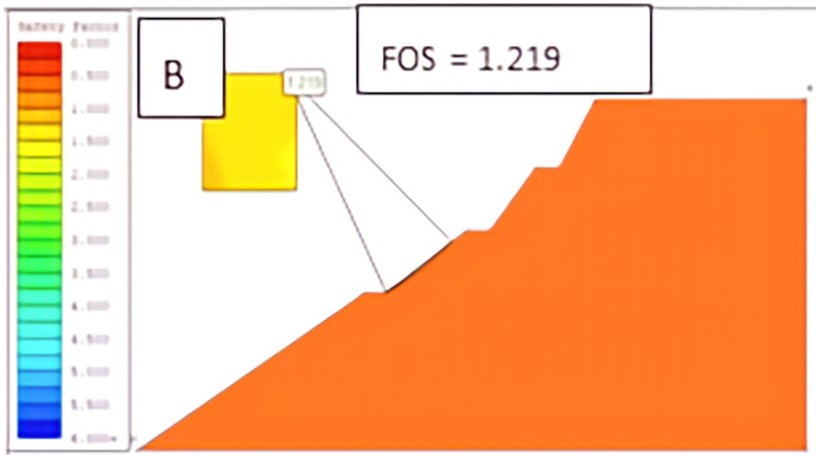

FOS = 1.219

**Fig 6. Effect of bench number on a factor of safety (D1S3).**

slope angle 45˚, increased to 0.77 at 35˚; it increased again to 0.89 at 28˚, 1.022 at 25˚, as an angle of the slope made 18˚ it reaches 1.151 which in turn made the slope section stable. The increase in the (H/V) ratio due to the flattening of the slope is also shown below. Here, flattening was conducted on the bottom slope of the section. The Spencer method calculated the safety factor as the former analysis.

Based on static dry conditions and a factor of safety of 0.545, the effect of slope angle was evaluated in the soil slope section (D2S1). According to the analysis shown below (Fig 8), the factor of safety, which was initially 0.545 at slope angle 46˚ increased to 0.684 at 35˚ it, increased again to 0.920 at 25˚ then 1.020 at 23˚, as an angle of the slope made 20˚ the factor of safety reaches 1.315 which in turn made the slope section stable. The increase in the (H/V) ratio due to the flattening of the slope is also shown below. Flattening was conducted here on the section's bottom slope (Fig 8). The Spencer method calculated the safety factor, similar to the former analysis.

**3.4.3 Vegetation cover.** Vegetation cover mitigates soil erosion by reducing its susceptibility. In addition to protecting the soil from the impact of raindrops and splashes, it also

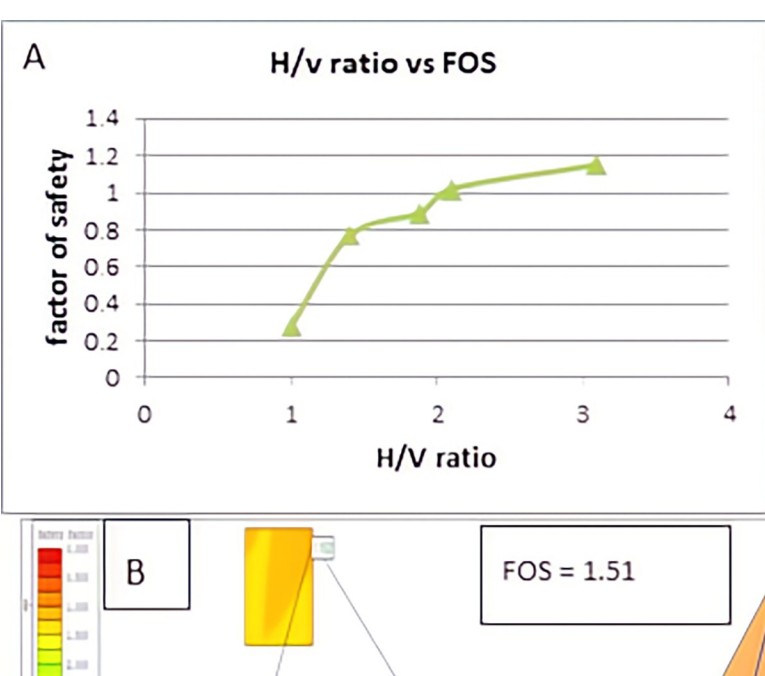

| Slope angle | slope ratio | H/V Ratio | FOS in slide 2D |
|---|---|---|---|
| tan (45) | 1H:1V | 1 | 0.240 |
| tan (35) | 1.5H:1V | 1.4 | 0.77 |
| tan (28) | 2H:1V | 1.88 | 0.89 |
| tan (25) | 2.5H:1V | 2.1 | 1.022 |
| tan (18) | 3H:1V | 3.09 | 1.151 |

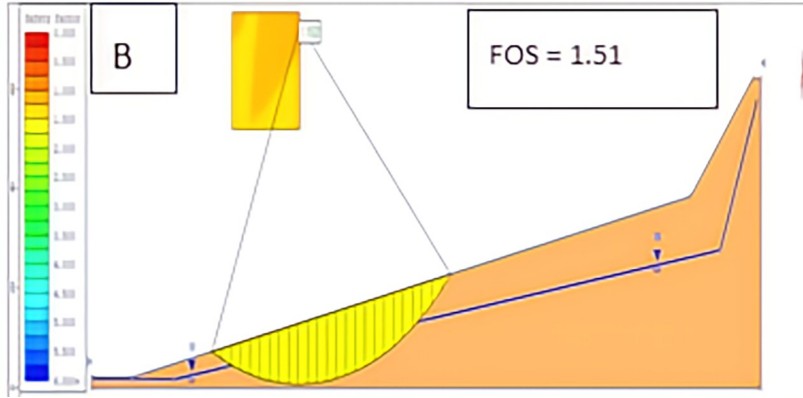

**Fig 7. The effect of slope flattening on the factor of safety (D1S2).**

minimizes the erosive potential of surface runoff and facilitates the infiltration of excess surface water. This is arguably one of the most effective methods of slope protection, particularly in preventing soil erosion on slopes. Covering the slope with grass prevents water from penetrating the surface materials. A practical and cost-effective method of stabilizing slopes is to cover them with a grass mat.

**3.4.4 Drainage systems.** The saturation of the subsoil and the subsequent build-up of pore water pressure increases the risk of slope failure. Enhancing slope stability by implementing an effective drainage system that will reduce pore water pressure and subsoil saturation is possible. This slope stabilization method requires regular surface and subsurface drain maintenance. By implementing a well-designed surface and subsurface drainage system, it is possible to minimize the build-up of pore water pressure within the subsoil of a slope, thereby enhancing its stability.

**3.4.5 Increasing public awareness.** In order to increase public awareness, individuals residing in or near areas with unstable slopes should be educated. The objective is to increase their understanding of the potential risks associated with slope failure and discourage everyday activities that could contribute to slope instability. These activities include tree cutting, overgrazing, construction of housing terraces, intensive agricultural practices, and inadequate drainage systems. To emphasize the importance of educating the local population about slope failure and its potential consequences for both human and property lives, it is crucial to emphasize the importance of educating the local people about the gravity of slope failure.

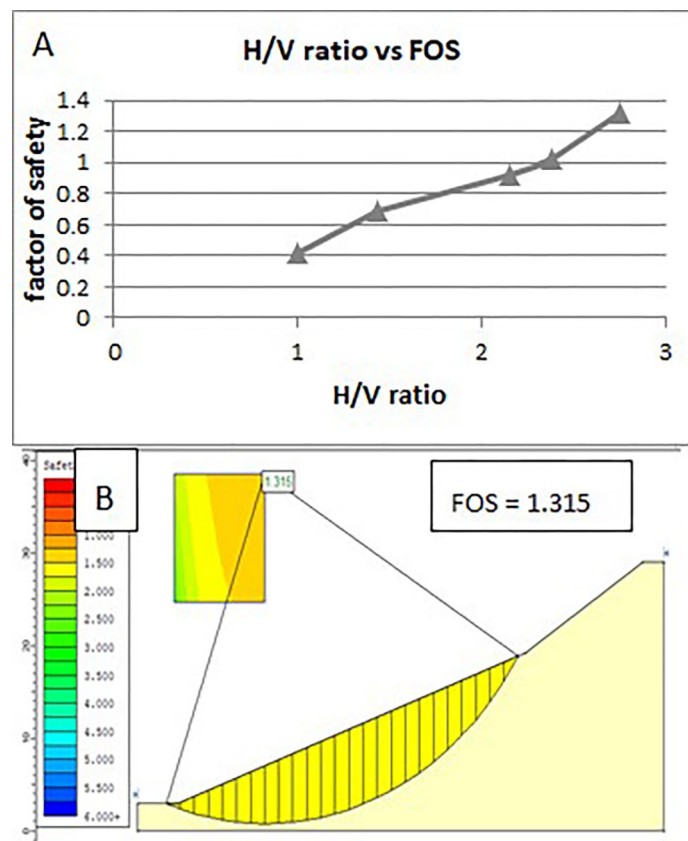

| slope angle | slope ratio | H/V Ratio | FOS in slide 2D |
|---|---|---|---|
| tan (46) | 1H:1V | 1 | 0.545 |
| tan (35) | 1.5H:1V | 1.43 | 0.684 |
| tan (25) | 2H:1V | 2.15 | 0.92 |
| tan (23) | 2.5H:1V | 2.38 | 1.02 |
| tan (20) | 3H:1V | 2.747 | 1.315 |

**Fig 8. Effect of slope flattening on the factor of safety (D2S1).**

## 4. Conclusion

A critical slope section along the Gerese to Belta road in Southern Ethiopia was investigated. To achieve this, fieldwork and laboratory testing were combined to assess the slope's stability. Field discontinuity surveys were conducted, and soil samples were collected for laboratory analysis. In the laboratory, soil's unit weight was determined through a direct shear test. In the stability analysis of slope sections, limit equilibrium and finite element methods were used. Different conditions were assessed, including dry and wet scenarios and static and dynamic loading. Based on field manifestations, such as slope toe removal, tilting slope face, and discontinuity orientation, three critical soil slopes were identified. Three soil slope sections were evaluated using finite element and limit equilibrium methods.

Furthermore, this study includes the evaluation of some comparative studies for a suggestion of remedial measures and modification of slope geometry for unstable critical soil slope sections (D1S2), (D1S3), and (D2S1) under different anticipated conditions. These slope sections were examined by benching (D1S3) at the dry, static state, and flattening of slope angle (D1S2 & D2S1) at Dynamic wet and Static dry conditions, respectively, under Slide 2D, and the following conclusion was made.

According to the analysis, the slope section of (D1S3) became stable with a safety factor of 1.222 due to the 5m wide benches used under slide 2D, which was initially unstable with a safety factor of 0.267. It was stabilized by a safety factor of 1.219 when three benches of the same width were used under slide 2. At the slope section (D1S2), reducing the slope angle or flattening from initial 45° to 35°, 28°, 25°, 18° increases the slope's safety factor from initial

0.240 to 0.77, 0.89, 1.022 1.151 respectively. Finally, at 18˚, the slope section became stable according to slide 2D analysis. The third slope section (D2S1) was flattened from an initial 46˚ to 35˚, 25˚ 23˚, and 20˚ then the factor of safety, in turn, increased from an initial 0.545 to 0.684, 0.920, 1.02, and 1.315 respectively. Finally, 20˚ of the slope section is stable according to slide 2D analysis.

## 4.1 Recommendation

This study analyzed slope stability along the Gerese to Belta road using limit equilibrium and finite element methods. A comparative analysis was conducted after the study, and several remedial measures were proposed, including benching and flattening critical slopes. These are the recommendations we came up with based on the research:

➢ It's best to stabilize soil slope section D1S3 using two benches, each with a width of 5 meters, or three benches, each with a width of 3 meters, based on the comparative analysis with Slide 2D.

➢ Future research should explore using appropriate software to stabilize rock slope sections.

➢ A surface and subsurface drainage system is strongly recommended to address stability concerns in other slope sections.

## Author Contributions

**Conceptualization:** Daniel Gebreyohannes, Ephrem Getahun.

**Data curation:** Daniel Gebreyohannes, Ephrem Getahun.

**Writing – original draft:** Daniel Gebreyohannes, Ephrem Getahun, Muralitharan Jothimani.

**Writing – review & editing:** Ephrem Getahun, Muralitharan Jothimani.

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
