## [Decision Letter · Decision Letter 0]

8 Nov 2023

PONE-D-23-31776Slope Stability Assessment in the Seismically and Landslide-Prone Road Segment of Gerese to Belta, Rift Valley, EthiopiaPLOS ONE

Dear Dr. jothimani,

Thank you for submitting your manuscript to PLOS ONE. After careful consideration, we feel that it has merit but does not fully meet PLOS ONE’s publication criteria as it currently stands. Therefore, we invite you to submit a revised version of the manuscript that addresses the points raised during the review process.

Please consider all comments Please submit your revised manuscript by Dec 23 2023 11:59PM. If you will need more time than this to complete your revisions, please reply to this message or contact the journal office at plosone@plos.org. Please include the following items when submitting your revised manuscript:A rebuttal letter that responds to each point raised by the academic editor and reviewer(s). You should upload this letter as a separate file labeled 'Response to Reviewers'.A marked-up copy of your manuscript that highlights changes made to the original version. You should upload this as a separate file labeled 'Revised Manuscript with Track Changes'.An unmarked version of your revised paper without tracked changes. You should upload this as a separate file labeled 'Manuscript'.

We look forward to receiving your revised manuscript.

Kind regards,

Ahmed Mancy Mosa, Ph.D.

Academic Editor

PLOS ONE

4. We note that Figure(s) 3, 7, 8, 9, 10, 11, 12, 13b, 14b, 15b and 16b in your submission contain copyrighted images. All PLOS content is published under the Creative Commons Attribution License (CC BY 4.0), which means that the manuscript, images, and Supporting Information files will be freely available online, and any third party is permitted to access, download, copy, distribute, and use these materials in any way, even commercially, with proper attribution. For more information, see our copyright guidelines: http://journals.plos.org/plosone/s/licenses-and-copyright.

a. You may seek permission from the original copyright holder of Figure(s) 3, 7, 8, 9, 10, 11, 12, 13b, 14b, 15b and 16b to publish the content specifically under the CC BY 4.0 license. 

5. We note that Figure 1 and 2 in your submission contain [map/satellite] images which may be copyrighted. All PLOS content is published under the Creative Commons Attribution License (CC BY 4.0), which means that the manuscript, images, and Supporting Information files will be freely available online, and any third party is permitted to access, download, copy, distribute, and use these materials in any way, even commercially, with proper attribution. For these reasons, we cannot publish previously copyrighted maps or satellite images created using proprietary data, such as Google software (Google Maps, Street View, and Earth). For more information, see our copyright guidelines: http://journals.plos.org/plosone/s/licenses-and-copyright.

a. You may seek permission from the original copyright holder of Figure 1 and 2 to publish the content specifically under the CC BY 4.0 license.  

Reviewers' comments:

Reviewer's Responses to Questions

**Comments to the Author**

1. Is the manuscript technically sound, and do the data support the conclusions?

Reviewer #1: Partly

Reviewer #2: Yes

2. Has the statistical analysis been performed appropriately and rigorously? 

Reviewer #1: N/A

Reviewer #2: Yes

3. Have the authors made all data underlying the findings in their manuscript fully available?

Reviewer #1: Yes

Reviewer #2: Yes

4. Is the manuscript presented in an intelligible fashion and written in standard English?

Reviewer #1: No

Reviewer #2: Yes

5. Review Comments to the Author

Reviewer #1: 1. The Abstract should be concise to highlight the main subject matter and aspect of the paper with brief findings.

2. In the literature review portion, several important and recent relevant contributions are missing.

3. In the Materials and methods portion, it is lack of results of lab test and field survey .

4. In the slope stability analysis portion, the content should be reorganized to support the research.

5. In the Remedial measure portion, there are no effective solutions for this problem.

6. The conclusion should be concise with focus on the primary research findings.

7. The manuscript should have written in a academic paper style.

Reviewer #2: The paper performs studies on slope stability assessment in the seismically and landslide-prone road segment of gerese to belta, rift valley, Ethiopia, in which tests and numerical analysis were done. The contents are interesting. The paper can be accepted for publication only if the following comments are addressed.

(1) When performing the shear tests, how do you predict the parameters are applicable to the slope analysis?

(2) In 2.4.2.4 Limit Equilibrium Method, how do you obtain the effective strength parameters?

(3) In 3.1, How do you consider the seismic effects?

(4)Do you consider the influence of rain infiltration?

6. PLOS authors have the option to publish the peer review history of their article (what does this mean?). If published, this will include your full peer review and any attached files.

Reviewer #1: **Yes: **JUN LIN

Reviewer #2: No

---

## [Author Response · Author response to Decision Letter 0]

19 Nov 2023

Dear Editor-in-Chief, PLOS ONE  

I would like to thank you for allowing me to revise my manuscript " Slope Stability Assessment in the Seismically and Landslide-Prone Road Segment of Gerese to Belta, Rift Valley, Ethiopia" PONE-D-23-31776. I am grateful for the insightful feedback provided by the reviewers and the opportunity to address their comments and suggestions. I would like to express my sincere appreciation to the reviewers for their valuable input, which has significantly enhanced the quality and clarity of my work. Their constructive comments have enabled me to refine my research and strengthen the overall contribution of the article. 

I have carefully considered the reviewer's suggestions and made the necessary revisions to address their concerns.

I am herewith providing a point-by-point response to each reviewer's comment.

Academic Editor Comments

In your Methods section, please provide additional information regarding the permits you obtained for the work. Please ensure you have included the full name of the authority that approved the field site access and, if no permits were required, a brief statement explaining why.

Author’s response: The current research was carried out during the ongoing road construction for public use, eliminating the necessity to obtain permission for collecting soil and rock samples in that area.

In your Data Availability statement, you have not specified where the minimal data set underlying the results described in your manuscript can be found.

Author’s response: All field and laboratory data utilized in this study have been incorporated within the paper, and a Data Availability Statement has been included in the revised version of the paper.

We note that Figure(s) 3, 7, 8, 9, 10, 11, 12, 13b, 14b, 15b and 16b in your submission contain copyrighted images.

Author’s response: The aforementioned figures were omitted in the revised paper.

We note that Figure 1 and 2 in your submission contain [map/satellite] images which may be copyrighted.

Author’s response: The authors created Figure 1. And citation mentioned for figure 2. 

Reviewer #1:

1. The Abstract should be concise to highlight the main subject matter and aspect of the paper with brief findings.

Author’s response: As per your instructions, the abstract has been rewritten and updated in the revised paper.

2. In the literature review portion, several important and recent relevant contributions are missing.

Author’s response: Updated the revised paper with additional (references 30 to 37), pertinent, and recent references.

3. In the Materials and methods portion, it is lack of results of lab test and field survey.

Author’s response: The laboratory test outcomes and field survey findings are displayed in subsections 2.4.2 and the annexure within the revised paper.

4. In the slope stability analysis portion, the content should be reorganized to support the research.

Author’s response: In the updated paper, elucidate the research findings within subsections 3.2.3 and 3.2.4.

5. In the Remedial measure portion, there are no effective solutions for this problem.

Author’s response: In the updated paper, elucidate the distinct remedial measures within subsections 3.4.1, 3.4.2, 3.4.3, 3.4.4, and 3.4.5.

6. The conclusion should be concise with focus on the primary research findings.

Author’s response: Re written concisely in the revised paper. 

7. The manuscript should have written in a academic paper style.

Author’s response: The manuscript underwent revisions in response to the reviewer's comments.

Reviewer #2: The paper performs studies on slope stability assessment in the seismically and landslide-prone road segment of gerese to belta, rift valley, Ethiopia, in which tests and numerical analysis were done. The contents are interesting. The paper can be accepted for publication only if the following comments are addressed.

(1) When performing the shear tests, how do you predict the parameters are applicable to the slope analysis?

Author’s response: During shear tests, parameters such as shear strength, soil properties, and results from laboratory tests are examined to understand their relevance to slope analysis.

(2) In 2.4.2.4 Limit Equilibrium Method, how do you obtain the effective strength parameters?

Author’s response: In the Limit Equilibrium Method used for slope stability analysis, effective strength parameters are typically obtained through laboratory testing and field investigations. The effective strength parameters crucial for this method include the cohesion (c) and the friction angle (φ) of the soil or rock mass

(3) In 3.1, How do you consider the seismic effects?

Author’s response: In the Ethiopian Rift Valley, considering seismic effects in slope stability analysis involves assessing the region's seismic hazard by studying historical seismic data and local ground motion parameters. Understanding the dynamic properties of soils and rocks, along with the geological conditions of slopes, is crucial.

(4) Do you consider the influence of rain infiltration?

Author’s response: Yes considered. 

Once again, I would like to express my gratitude for the reviewers' insightful comments and the opportunity to revise and resubmit my work. I sincerely hope you and the editorial board will find the revised manuscript satisfactory for publication in the Egyptian Journal of Remote Sensing and Space Sciences. Thank you for considering this revised submission. I look forward to hearing from you regarding the outcome of the review process.

---

## [Decision Letter · Decision Letter 1]

19 Dec 2023

Slope Stability Assessment in the Seismically and Landslide-Prone Road Segment of Gerese to Belta, Rift Valley, Ethiopia

PONE-D-23-31776R1

Dear Dr. jothimani,

We’re pleased to inform you that your manuscript has been judged scientifically suitable for publication and will be formally accepted for publication once it meets all outstanding technical requirements.

Kind regards,

Ahmed Mancy Mosa, Ph.D.

Academic Editor

PLOS ONE

Additional Editor Comments (optional):

Reviewers' comments:

Reviewer's Responses to Questions

**Comments to the Author**

1. If the authors have adequately addressed your comments raised in a previous round of review and you feel that this manuscript is now acceptable for publication, you may indicate that here to bypass the “Comments to the Author” section, enter your conflict of interest statement in the “Confidential to Editor” section, and submit your "Accept" recommendation.

Reviewer #1: All comments have been addressed

2. Is the manuscript technically sound, and do the data support the conclusions?

Reviewer #1: Yes

3. Has the statistical analysis been performed appropriately and rigorously? 

Reviewer #1: Yes

4. Have the authors made all data underlying the findings in their manuscript fully available?

Reviewer #1: Yes

5. Is the manuscript presented in an intelligible fashion and written in standard English?

Reviewer #1: Yes

6. Review Comments to the Author

Reviewer #1: The manuscript has been revised completely in accordance with the raised comments. The manuscript discussed a slope stability problem on several sections of the Gerese-Belta route in Southern Ethiopia, and a lot of fieldwork, laboratory testing and numerical analysis have performed well on the slope stability. The manuscript could be accepted and published in this journal.

7. PLOS authors have the option to publish the peer review history of their article (what does this mean?). If published, this will include your full peer review and any attached files.

Reviewer #1: **Yes: **JUN LIN

---

## [Editor Report · Acceptance letter]

30 Jan 2024

PONE-D-23-31776R1 

PLOS ONE

Dear Dr. Jothimani, 

I'm pleased to inform you that your manuscript has been deemed suitable for publication in PLOS ONE. Congratulations! Your manuscript is now being handed over to our production team.

Kind regards, 

on behalf of

Dr. Ahmed Mancy Mosa 

Academic Editor

PLOS ONE